# Viral Envelope Evolution in Simian–HIV-Infected Neonate and Adult-Dam Pairs of Rhesus Macaques

**DOI:** 10.3390/v16071014

**Published:** 2024-06-25

**Authors:** Elena E. Giorgi, Hui Li, Bhavna Hora, George M. Shaw, Kshitij Wagh, Wilton B. Williams

**Affiliations:** 1Fred Hutchinson Cancer Center, Seattle, WA 98109, USA; 2Departments of Medicine and Microbiology, Perelman School of Medicine, University of Pennsylvania, Philadelphia, PA 19104, USA; huili2@pennmedicine.upenn.edu (H.L.); shawg@upenn.edu (G.M.S.); 3Duke Human Vaccine Institute, Duke University School of Medicine, Durham, NC 27710, USA; bhavna.hora@duke.edu; 4Los Alamos National Laboratory, Los Alamos, NM 87544, USA; kshitij.wagh@duke.edu; 5Department of Surgery, Duke University School of Medicine, Durham, NC 27710, USA; 6Department of Integrative Immunobiology, Duke University School of Medicine, Durham, NC 27710, USA

**Keywords:** viral Env evolution, dam–neonate infection pairs, macaque SHIV infection, neonatal SHIV immunity, antibody–virus coevolution, NAb selection pressures on HIV-1 Env

## Abstract

We recently demonstrated that Simian–HIV (SHIV)-infected neonate rhesus macaques (RMs) generated heterologous HIV-1 neutralizing antibodies (NAbs) with broadly-NAb (bNAb) characteristics at a higher frequency compared with their corresponding dam. Here, we characterized genetic diversity in Env sequences from four neonate or adult/dam RM pairs: in two pairs, neonate and dam RMs made heterologous HIV-1 NAbs; in one pair, neither the neonate nor the dam made heterologous HIV-1 NAbs; and in another pair, only the neonate made heterologous HIV-1 NAbs. Phylogenetic and sequence diversity analyses of longitudinal Envs revealed that a higher genetic diversity, within the host and away from the infecting SHIV strain, was correlated with heterologous HIV-1 NAb development. We identified 22 Env variable sites, of which 9 were associated with heterologous HIV-1 NAb development; 3/9 sites had mutations previously linked to HIV-1 Env bNAb development. These data suggested that viral diversity drives heterologous HIV-1 NAb development, and the faster accumulation of viral diversity in neonate RMs may be a potential mechanism underlying bNAb induction in pediatric populations. Moreover, these data may inform candidate Env immunogens to guide precursor B cells to bNAb status via vaccination by the Env-based selection of bNAb lineage members with the appropriate mutations associated with neutralization breadth.

## 1. Introduction

The envelope (Env) trimer on the surface of human immunodeficiency virus type 1 (HIV-1) mediates fusion-dependent entry into CD4+ target cells [1,2] and thus is a primary target for antibodies to prevent infection [2,3,4,5]. However, the HIV-1 Env undergoes tremendous sequence diversification during infection: a hallmark of HIV-1 immune evasion [6,7,8,9]. In particular, mutations or genetic sequence changes in the Env sites targeted by antibodies that can block viral entry and neutralize HIV-1 have been reported to be the viral strategy to elude neutralization [6,7,10,11,12]. Env mutations that evade select antibodies may also increase susceptibility to other antibody lineages in a phenomenon reported as the cooperation of B cell lineages in chronic HIV-1 infection [8]. This phenomenon was defined in studies of viral Env and antibody coevolution in chronic HIV-1 infection [6,8] and supports the notion that Env variants, during viral infection with mutations that confer enhanced neutralization by members of an antibody lineage, may serve as immunogens to mature such lineages to enhanced neutralization potency and/or breadth against different HIV-1 strains [6].

Broadly neutralizing antibodies (bNAbs) target common epitopes on geographically diverse strains of the HIV-1 Env, but have only been found in a subset of people living with HIV-1 (PLWH) [5,13]. To date, the induction of HIV-1 Env bNAbs is the goal of an efficacious HIV-1 vaccine, which has been elusive [3,14,15]. In bNAb lineage vaccine design, priming immunogens trigger the activation of precursor bNAb B cells and boosting immunogens, mature members of the B cell lineage, to bNAb status [14,15,16]. Appropriate boosting immunogens must select for intermediate bNAb lineage members with rare or improbable mutations that are associated with a heterologous HIV-1 neutralization capacity during the maturation from precursor to bNAb status [17]. Hence, boosting immunogens must encode residues designed to select more mutated intermediate bNAb lineage members in an attempt to recapitulate the Env diversity associated with the development of heterologous HIV-1 neutralization [7,11]. The success of this vaccine strategy relies on the isolation of monoclonal bNAbs and computational inference of the unmutated common ancestor (UCA) and intermediate antibodies in the bNAb lineage; the identification of Envs during viral evolution when generated as recombinant proteins demonstrated an increased affinity for binding antibodies at these different stages of bNAb lineage maturation [16]. Thus, antibody and virus coevolution studies in PLWH who generated bNAbs facilitated the identification of candidate priming and boosting immunogens that may recapitulate the development of bNAb lineages via vaccination [6,8,9,18]. Recently developed Simian–HIV (SHIV)-bearing transmitted–founder Envs associated with bNAb induction in humans [19,20] have provided the opportunity to study bNAb induction in both adult [11] and pediatric [12] models of rhesus macaques (RMs), where we may also potentially identify candidate Envs for bNAb lineage maturation via vaccination.

HIV-1 immune evasion is facilitated by at least three interrelated Env properties that act in concert to preserve the Env structure and remain functional for cell entry: glycan shielding [7,10], epitope variation [21], and conformational masking [22]. We characterized these facets of Env evolution using longitudinal sequences by studying amino acid evolution [11,23], glycan shield evolution [7], and hypervariable loop evolution and recombination [24] in a subset of neonate and adult/dam pairs of RMs. While all animals were infected with the same pathogenic SHIV, neonate RMs generated heterologous HIV-1 NAbs with characteristics of bNAbs (Holmes S, Williams WB et al., submitted) at a higher frequency compared to paired adult RMs. Here, we identified mutation signatures in longitudinal HIV-1 Envs, including in sites targeted by bNAbs that were associated with the development of neutralization breadth by heterologous HIV-1 NAbs in neonatal SHIV infection of RMs. 

## 2. Materials and Methods

### 2.1. Experimental Model

Viral Env sequences were previously obtained from SHIV CH848 10.17 N133DN138T.E169K-infected Indian RMs (Macaca mulatta) that were housed indoors at BIOQUAL, Inc., Rockville, MD, USA and were maintained in accordance with the Association for Assessment and Accreditation of Laboratory Animals with the approval of the IACUC as described [12] (Holmes S, Williams WB et al., submitted). The research was conducted in compliance with the Animal Welfare Act and other federal statutes and regulations relating to animals and experiments involving animals and adhered to the principles stated in the Guide for the Care and Use of Laboratory Animals, NRC Publication, 2011 edition.

Our cohort of four neonate and dam RM pairs were as follows: V058 (neonate)/BD62 (adult); V093 (neonate)/V060 (adult); V096 (neonate)/V061 (adult); and V059 (neonate)/BJ56 (adult). Neonate RMs V058 and V096 were males, whereas neonate RMs V093 and V059 were females, and all adult RMs were females. All dams were intravenously challenged with SHIV subsequent to dam rearing to prevent SHIV antigen pre-exposure in the neonate RMs. The animal protocols for viral challenges and blood draws were previously described [12] (Holmes S., Williams W.B. et al., submitted). Holmes S. et al. (submitted) extensively described a total of 11 neonate/dam pairs for the comparison of virologic and immunologic responses to the same SHIV. The 4 neonate/dam pairs described in this manuscript included V093/V060 and V058/B62 (Group 1) as the only neonate/dam pairs in which both neonates and dams generated heterologous HIV-1 NAbs, whereas V096/V061 (Group 2) was representative of 5 neonate/dam pairs in which only the neonate generated heterologous HIV-1 NAbs, and V059/BJ56 (Group 3) was representative of 4 neonate/dam pairs in which neither the neonate nor the dam generated heterologous HIV-1 NAbs (Figure 1). The representative neonate/dam pairs had the most viral Env sequences recovered from longitudinal time points.

### 2.2. Viral Envelope Sequencing

Viral Env sequencing was achieved via the amplification of the SHIV 3′ half genomes from the plasma of RMs using the single-genome sequencing (SGS) method [12,20]. We were limited to the successful amplification of viral cDNA from plasma samples where the viral load was ≥3 Logs of viral RNA copies/mL.

### 2.3. Phylogenetic Analyses

The phylogenetic reconstruction of 758 longitudinally sampled amino acid Env sequences from 4 RM neonates and 4 RM paired dams, rooted on the CH848 10.17 DT.E169K SHIV-infecting strain, was conducted using FastTree software [26] with the JTT (Jones–Taylor–Thornton 1992) model and CAT approximation. Sequence trees from individual neonate/adult-dam pairs were made using IQ-TREE software [27] with the HIVb model [28] and ultrafast bootstrap for branch support. All trees were then rendered and color coded using the R statistical software platform version 4.3.0 (21 April 2023) [http://www.R-project.org] with the packages ape and phytools.

### 2.4. Sequence Diversity

Within each time point, pairwise Hamming distances, and the Hamming distances from the CH848 10.17 DT.E169K Env were calculated, plotted, and statistically compared using the R statistical software platform [http://www.R-project.org]. Common sites of variability across animals were identified using the previously described pipeline “Longitudinal Antigenic Sequences and Sites from Intra-Host Evolution” or LASSIE [23]. Briefly, for each animal, we first identified all sites at which the frequency of the corresponding residue in the SHIV challenge CH848 10.17 DT.E169K was less than 50% within at least one longitudinal time point. We then ranked all sites based on how many times they were shared across multiple animals. Logo plots were created using the LANL tool AnalyzeAlign [https://www.hiv.lanl.gov] and Pixel plots using the LANL tool Pixel [https://www.hiv.lanl.gov/content/sequence/pixel/pixel.html]. Differences in the mean Hamming distances were tested using two-sided Wilcoxon tests. All statistical tests, graphs, and coding were performed on the R platform [http://www.R-project.org].

## 3. Results

### 3.1. Cohort of SHIV-Infected RMs

We recently described plasma antibody responses in 11 pairs of neonate and adult-dam RMs infected with a highly infectious and pathogenic SHIV bearing an HIV-1 Env CH848 10.17 DT.E169K that was sensitive to neutralization by bnAb lineages targeting the HIV-1 Env V2 and V3 regions (Holmes S., Williams W.B. et al., submitted). We previously defined the induction of plasma HIV-1 NAbs against difficult-to-neutralize or tier 2 heterologous HIV-1 strains in neonatal SHIV infection as a marker for bNAb induction [12] (Holmes S, Williams WB et al., submitted). To investigate the mechanisms underlying neonatal bNAb induction, here, we selected four pairs of neonate and dam RMs representing three different scenarios (Figure 1): (i) both the neonate and dam developed plasma heterologous HIV-1 NAbs, Group 1: V093 (neonate)/V060 (dam) and V058 (neonate)/BD62 (dam) pairs; (ii) the neonate developed high-titer plasma heterologous HIV-1 NAbs but not the dam, Group2: V096 (neonate)/V061 (dam) pair; and (iii) neither the neonate nor the dam developed plasma heterologous HIV-1 NAbs, Group 3: V059 (neonate)/BJ56 (adult) pair. Figure 1 shows the neutralization titers against four tier 2 or difficult-to-neutralize HIV-1 strains tested using plasma from all eight RMs. In Group 1 animals, where both neonates and dams generated heterologous HIV-1 NAbs (Figure 1A,B), and Group 2 animals, where only the neonate generated heterologous HIV-1 NAbs (Figure 1C), we detected consistent neutralization against heterologous tier 2 HIV-1 strains tested at ≥2 time points. These data supported the categorization of the three groups of RMs described above in this study. A more comprehensive profile of neutralization titers against a larger panel of HIV-1 strains tested using plasma from these animals was previously described [12] (Holmes S, Williams WB et al., submitted).

### 3.2. Phylogenetic Analyses of Longitudinal HIV-1 Envs in SHIV-Infected RMs

A total of 758 longitudinally sampled Env sequences were obtained from eight animals across different time points; 454 Env sequences were from neonate RMs sampled at months (M) 06, 12, 18, and 24, and 304 Env sequences were from paired adult RMs sampled at M04, 06, 16, 14, and 18 across animals. For our analyses, we were limited to time points where the plasma viral loads in the animals were ≥3 Logs, from which Env sequences could be successfully recovered as described [12] (Holmes S, Williams W.B. et al., submitted).

The phylogenetic reconstruction of all sampled Env sequences rooted on Env CH848 10.17 DT.E169K showed that the neonate and dam sequences tended to segregate in the overall tree (Figure 2). The two neonates in Group 1 that developed heterologous HIV-1 NAbs (V093 and V058) formed their own subclades. The third neonate that developed heterologous HIV-1 NAbs, V096, clustered together with the neonate that did not develop heterologous HIV-1 NAbs, V059, at the first time point, but V096 sequences from later time points formed a subsequent subclade. A similar pattern was observed in the adult dams: the two dams that developed heterologous HIV-1 NAbs (V060, M12 and M16 sequences; and BD62, M06 sequences) clustered together, with the exception of the first time point from dam V060, which clustered with the sequences from the other two dams (BJ56 and V061) that did not develop heterologous HIV-1 NAb responses. Both the adult-dam V060 and the neonate V096 had no evidence of heterologous HIV-1 NAbs at the first sampled time point (Figure 1). Thus, overall, we observed that sequences from animals at the time points when heterologous HIV-1 NAbs were detected clustered together on tree subclades farther away from the CH848 10.17 DT.E169K tree root, whereas sequences from animals at the time points when heterologous HIV-1 NAbs were not detected (including the first time point from animals V060 and V096, who developed heterologous HIV-1 NAbs at later time points) clustered together on the tree subclade closest to the CH848 10.17 DT.E169K tree root. These data suggested that Envs from RMs that generated heterologous HIV-1 NAbs had more sequence similarity than those from animals that did not develop heterologous HIV-1 NAbs.

Env sequences from heterologous HIV-1 NAb-developing neonates V058 and V096 formed their own tree subclades, whereas those from heterologous HIV-1 NAb-developing neonate V093 formed a subclade branching out of heterologous HIV-1 NAb-developing dams V060 and BD62. By the latest time point sampled, M24, the three neonates (V093, V058, and V096) that developed heterologous HIV-1 NAb responses had the longest branch lengths from the CH848.10.17 DT.E169K tree root compared to all other animals (FDR-adjusted *p* < 2 × 10^−4^ using the Wilcoxon test; Figure 2B), with the sole exception of the heterologous HIV-1 NAb-developing dam BD62, who showed lower, but not significantly so, branch lengths as compared to V096.

Phylogenetic analyses of each individual pair of neonate and adult RMs revealed that in all four pairs, the Env sequences closest to the CH848.10.17 DT.E169K tree root were from the adult/dam, not the neonate (Figure 3). This was particularly striking for adult and neonate pairs BD62 andV058 (Figure 3A), and V060/V093 (Figure 3B), respectively, where we had Env sequences available for longitudinal time points spanning 18–24 months of SHIV infection. In all four phylogenetic trees (Figure 3A–D), neonate and paired dam sequences formed distinct subclades with no interleafing.

A single Env sequence from neonate V093 sampled at time point M06 was the sole exception, as it clustered with the paired dam’s (V060) Env sequences and away from all other V093 Env sequences. When compared in a diversity-sorted highlighter plot, this M06 V093 outlier sequence (V093_M06_3H_A08) was closest to three M12 dam sequences (5 amino acid differences) than its closest neonate sequences (11 amino acid differences) (Figure 4). The V093 and V060 sequences were generated in independent experiments, thus reducing the likelihood of cross-contamination between samples. Thus, these data raised the hypothesis that both neonate and dam pairs have similar Envs soon after infection that may diversify differently due to host-specific immune pressures.

### 3.3. Viral Diversification in SHIV-Infected RMs

Using phylogenetic analyses, we observed that Env sequences from animals that developed heterologous HIV-1 NAbs were the farthest away from the CH848.10.17 DT.E169K Env used to root the tree (Figure 2B). Since longer branch lengths indicate a higher sequence diversity, we explored whether viral diversity was a driver for heterologous HIV-1 NAb development. To this end, for each animal, at each time point, we computed two measures of viral genetic diversity. First, we calculated the Hamming distance (HD) distribution from the CH848 10.17 DT.E169K Env in the infecting SHIV, defined as, for each sequence at any given time point, the number of amino acid differences between the sequence and the infecting SHIV, divided by the full Env length. This first measure of viral diversity tracks how the viral population diverges from the initial founder virus. In addition, to measure whether the viral population evolved distinct sub-lineages once it started diversifying from that initial founder lineage, we calculated the pairwise HD distribution, defined as the number of amino acid differences between all sequence pairs within each time point divided by the full Env length.

By M24, all neonates that developed heterologous HIV-1 NAbs (V093, V058, and V096) had a median HD from the CH848 10.17 DT.E169K Env in the infecting SHIV of 6% or higher, which was significantly higher compared to the neonate that did not develop heterologous HIV-1 NAbs (V059); all comparisons had a *p* < 2.2 × 10^−16^ using the Wilcoxon test (Figure 5A). While we did not have available sequences at M24 for the adult RMs, a similar trend was observed among the adult RMs. Here, we found that the two adult RMs that developed heterologous HIV-1 NAbs (V060 and BD62) had significantly higher median pairwise HDs compared to the ones that did not (BJ56 and V061); all comparisons had a *p* < 2 × 10^−9^ using the Wilcoxon test at the latest common time point for which sequences were available: M12 for dams BD62, V060, and BJ56. The same level of significance was obtained when compared to dam V061 (*p* < 2 × 10^−9^ using the Wilcoxon test), though for this animal, the only time point at which sequences were available was M06. Thus, these data demonstrated that both neonate and dam RMs that developed heterologous HIV-1 NAbs had significantly higher median HDs from the CH848.10.17 DT.E169K Env in the infecting SHIV as compared to their counterparts that did not develop heterologous HIV-1 NAbs.

Within the two RM pairs where both the neonate and the dam developed heterologous HIV-1 NAbs (Group 1: V093/V060 and V058/BD62), by M18, both neonates had significantly higher median pairwise HDs compared to the respective paired dams; both comparisons showed a *p* < 2.4 × 10^−9^ using the Wilcoxon test (Figure 5A). Equivalent trends were observed for the pairwise HDs (Figure 5B): by M24, all neonates that developed heterologous HIV-1 NAbs had a median pairwise HD of 2% or higher, which was significantly higher than the neonates that did not (all comparisons had *p* < 2.2 × 10^−16^ using the Wilcoxon test). For the heterologous HIV-1 NAb-developing dams, while they had statistically significantly lower pairwise HDs at M6 and M18 compared to the paired neonates at M18 (*p* < 2.2 × 10^−16^ using the Wilcoxon test), they were still statistically significantly higher than the dams that did not develop heterologous HIV-1 NAbs (*p* < 2.2 × 10^−16^ using the Wilcoxon test) (Figure 5B). These data implicated differences in the magnitude and/or kinetics of diversification in Envs for neonates and dams that developed heterologous HIV-1 NAbs, as well as differences between neonates and adults among animals that develop heterologous HIV-1 NAbs.

### 3.4. Env Mutation Patterns Associated with the Development of Heterologous HIV-1 NAbs

Next, we compared the mutational patterns across RMs to identify specific signatures associated with the development of heterologous HIV-1 NAbs. We performed a longitudinal Env diversity analysis using the previously described Longitudinal Antigenic Sequences and Sites from Intra-Host Evolution (LASSIE) strategy, which identifies Env sites under putative immune selection pressure [23]. Overall, there were 22 sites in the Env, of which 2 were part of a deletion in the variable region V4, where a 50% or larger loss of the CH848 1017.DT.E169K residue was observed in three or more RMs (Figure 6A). Of these sites, the RMs that developed heterologous HIV-1 NAbs carried the most changes (14–17), whereas the RMs that did not develop heterologous HIV-1 NAbs carried nine or less changes, reflecting the fact that a higher Env diversity, indicative of enhanced immune selection pressure, is a driver for the earlier development of heterologous HIV-1 NAb responses.

At 11 of the 22 sites, we observed mutations shared by both RMs that developed heterologous HIV-1 NAbs, as well as those that did not (Figure 6A, residues labeled in green), including the acquisition of glycosylation sites N133, N138, and N230. However, there were an additional 9 sites associated with heterologous HIV-1 NAb development: at 5 of these sites, three or more out of the five animals that developed heterologous HIV-1 NAbs shared mutations (P11S, R59K, N187S, N362T/D/S, and N461D) not observed in any of the animals that did not develop heterologous HIV-1 NAbs (Figure 6A, residues labeled in red). At the other four sites we observed mutations (G139R, E269G, H289R, and K327R) that were also found in two of the animals that did not develop heterologous HIV-1 NAbs, dam-V061 and neonate-V059, but at a much lower frequency (<12%) than the frequency in the RMs that developed heterologous HIV-1 NAbs (Figure 6A). None of the mutations at variable sites, which were enriched in the animals that developed heterologous HIV-1 NAbs, were found in the adult RM BJ56 that did not develop heterologous HIV-1 NAbs (Figure 6B). The mutations R59K and K327R introduced highly conserved amino acids in the HIV-1 M-group with frequencies of 95.16% and 98.32%, respectively (www.hiv.lanl.gov), thus raising the possibility that the recapitulation of global HIV-1 diversity by the autologous Env quasispecies could be associated with breadth development.

As recently described [12] (Holmes S., Williams W.B. et al., submitted), by the last sampled time point, all RMs filled the N133 glycan hole, and seven out of eight filled the N138 glycan hole, both within the Env of the infecting SHIV. While all neonate RMs acquired an N230 glycan, of the four paired dams, only the two that developed heterologous HIV-1 NAbs, V060 and BD62, acquired it as well. Using analogous methods (i.e., LASSIE), in our recent study of SHIV CH848 10.17 DT.E169K-infected neonate RMs (Holmes S., Williams W.B. et al., submitted), we identified five Env mutations that were associated with breadth development, defined as plasma neutralization at M18 of ≥30% of heterologous HIV-1 strains in the virus panel: G139E, E153G, K327R, N362T/D, and R432K. Three of these five previously identified residues acquired via mutations, 139E, 327R, and 362T/D, were also identified in this study to be associated with heterologous HIV-1 NAb development when we compared neonate and adult-dam pairs of RMs. All RMs that developed heterologous HIV-1 NAbs by the last time point studied (M24 for the three neonates and M16 and M18, respectively, for the dams V060 and BD62) had a mutation that resulted in a 327R residue. Additionally, residues 362T/D were found in the two pairs of RMs where both the dam and neonate developed heterologous HIV-1 NAbs, but not in the pair where only the neonate developed heterologous HIV-1 NAbs (Figure 6A). Interestingly, a loss of a glycan or asparagine at position 362 has been found to be associated with V3-bnAb resistance in previous studies [29].

Env deletions in the hypervariable regions V1, V2, and V4 in particular have been previously reported in both adult and neonate infections by different SHIVs using variations of the CH848 Env; SHIV CH848TF [11] and SHIV CH848 10.17 DT.E169K [12] (Holmes S, Williams WB et al., submitted). Here, one deletion in V4, at sites 401 and 402, was shared by all RMs that generated heterologous HIV-1 NAbs and in only one of three RMs that did not generate heterologous HIV-1 NAbs. Of the RMs that generated heterologous HIV-1 NAbs, neonate V058 carried the deletion at M06 and M12, but at the subsequent time points, the deletion was filled with a glycan (Figure 6A). These data suggested that Insertions and Deletions (INDELS) may also contribute to sequence diversification in viral Env evolution in pediatric populations and impact the development of heterologous HIV-1 NAbs, as described in adult SHIV-infected RMs [11].

## 4. Discussion

Viral Env and bNAb coevolution studies in chronic HIV-1 infection [6,9,18] and macaque SHIV infection [11] have been seminal in the field of HIV-1 vaccinology for the development of an efficacious bNAb-inducing HIV-1 vaccine [16,30,31,32]. BNAb lineage vaccine design [16] and reverse vaccinology 2.0 [32] are two key concepts being applied in the field of HIV-1 vaccine development, and critical to both concepts is the identification of Envs as candidate immunogens capable of recapitulating bNAb B cell maturation via vaccination [14,15,30]. Thus, Env evolution during HIV-1 infection [6,9,18] and macaque SHIV infection [11,12] that elicits heterologous HIV-1 NAbs with characteristics of bNAbs can inform immunogens capable of recapitulating bNAb induction via vaccination. Here, we performed Env evolution studies in a limited number of neonate and adult-dam pairs infected with the same SHIV and demonstrated that Env sequence diversification, including candidate bNAb-inducing mutations, were associated with heterologous HIV-1 NAb induction in neonate RMs and with potentially faster kinetics than adult RMs.

There are at least four aspects of Env evolution that may facilitate immune evasion while preserving the properties of the Env trimer to mediate viral entry [10,21,22,33]. Our sequence analysis pipeline identified Env sites under selection pressure by investigating one of the four aspects of Env evolution: amino acid and glycan mutations using LASSIE [11,23]. The HIV-1 Env mutates to escape neutralization; thus, the identification of mutations in NAb-accessible epitopes on Env is suggestive of selection pressures by NAbs elicited by SHIV infection in RMs. In particular, mutations that filled glycan holes in the Env V1 region have been associated with selection pressures from autologous HIV-1 NAbs [11,12], in agreement with mutations that fill glycan-deficient holes in Env [7,34], whereas those in the ^324^GDIR^327^ motif may be linked with selection pressure from V3-glycan bnAbs [9]. Thus, Envs with bnAb selection pressures may be candidate Envs for selecting B cell lineages capable of maturing into bnAb status. Future studies will isolate antigen-reactive B cells with a heterologous HIV-1 neutralization capacity, and the Env sites identified in this study would be useful in designing candidate immunogens to mature these lineages to neutralization breadth via vaccination.

One limitation of this study is that we sequenced fewer time points from adult/dam RMs compared to neonate RMs due to very low viral loads in the adult RMs. However, a high plasma viral load has been associated with b/NAb induction [12,35,36], suggestive of persistent antigen selection of B cell receptors for affinity maturation and the development of b/NAbs. We recently found that a higher frequency of neonates compared to adults/dams generated heterologous HIV-1 NAbs, in agreement with neonates having a higher mean plasma viral load over time (Holmes S., Williams W.B. et al., submitted). In this study, the five animals that developed heterologous HIV-1 NAbs between M06 and M18 had a viral load consistently higher than the three animals that did not develop heterologous HIV-1 NAbs (Holmes S., Williams W.B. et al., submitted). We found that the neonate and adult/dam RMs that generated heterologous HIV-1 NAbs had a higher level of sequence diversity than RMs that did not generate heterologous HIV-1 NAbs. Moreover, the neonates that generated heterologous HIV-1 NAbs had more Env sequence diversification than the dams that also generated heterologous HIV-1 NAbs. Additionally, we identified signature sites where mutations were observed exclusively in the animals that developed heterologous HIV-1 NAbs, and three of these mutations we had previously associated with breadth development (Holmes S., Williams W.B. et al., submitted), suggesting that similar pathways of diversification are needed to go from the induction of heterologous HIV-1 NAbs to bNAb development with neutralization breadth. Altogether, these data support the hypothesis that Env sequence diversity is a key driver for the development of neutralization breadth [7,11]. Additionally, these data may inform a potential mechanism for neonatal immunity favorably generating heterologous HIV-1 NAbs compared to adults, as described [37,38] (Holmes S., Williams W.B. et al., submitted).

Previous studies have revealed both antibody [7,10,11,12] and CD8 T-cell [8] immune selection pressures on Envs during HIV-1 or macaque Simian–HIV (SHIV) infections, but our analyses were focused on antibody selection pressures. We identified candidate Env mutations associated with heterologous HIV-1 NAb induction in neonate RMs that may select B cell lineages for maturation to neutralization status, including bNAbs. Moreover, the results from our studies provide an opportunity to functionally characterize these mutations and define the B cell lineages that drive them. While proposals for childhood vaccination against HIV-1 have been put forth [39], our understanding of the immune selection pressures in the HIV-1 Env during infection in studies of adult populations have typically been used to inform viral pathogenesis and HIV-1 vaccine immunogen design. Thus, our data contribute to an important, understudied area of understanding the mechanisms of neonatal immunity that support heterologous HIV-1 NAb induction.

## Figures and Tables

**Figure 1 viruses-16-01014-f001:**
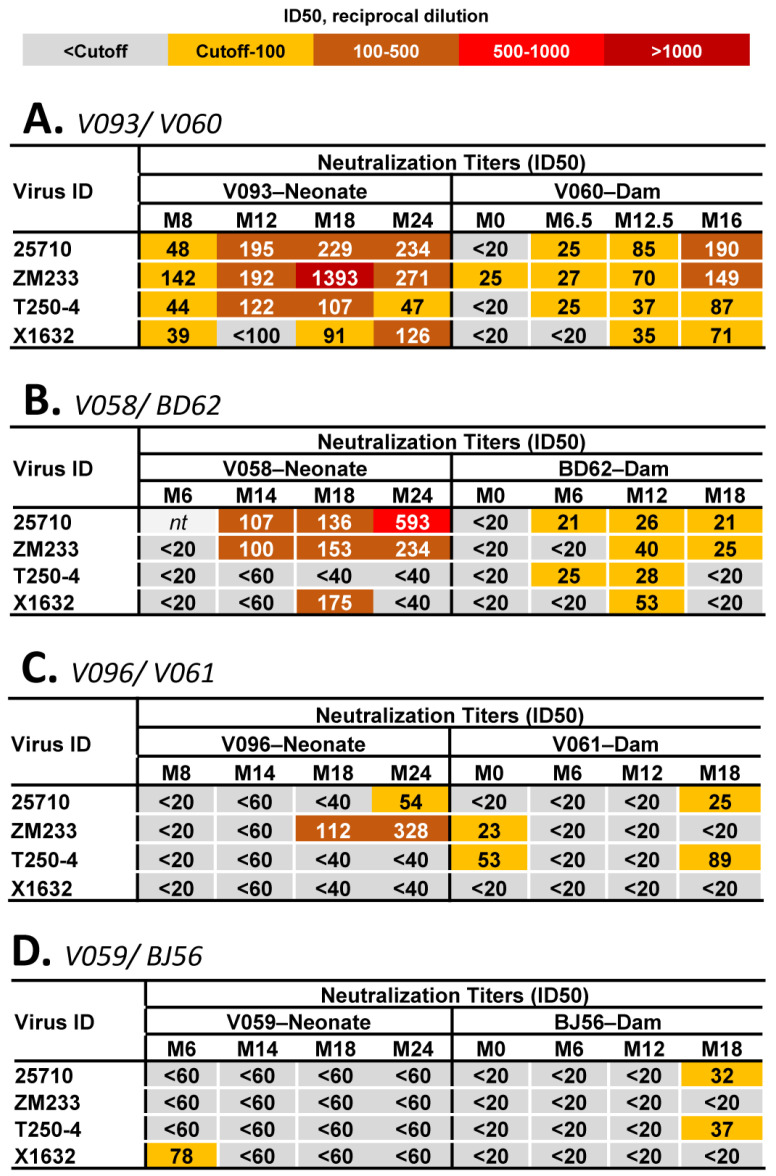
Heterologous HIV-1 neutralization profiles of plasma antibodies elicited by neonatal and adult SHIV infection. Longitudinal plasma antibodies spanning 24 months following neonatal and adult SHIV infection were tested for neutralization of heterologous HIV-1 reference strains from the global panel [25]. The neutralization titer was measured in TZM-bl cells and reported as ID50 (reciprocal dilution). Each heatmap shows the neutralization titers of plasma antibodies in the corresponding neonate and dam pairs: (**A**) V093 (neonate)/V060 (dam); (**B**) V058 (neonate)/BD62 (dam); (**C**) V096 (neonate)/V061 (dam); and (**D**) V059 (neonate)/BJ56 (dam). As shown by the key at the top of the figure, the heatmap is color coded to show the magnitude of NAb titers generated in each animal at different time points against each virus tested.

**Figure 2 viruses-16-01014-f002:**
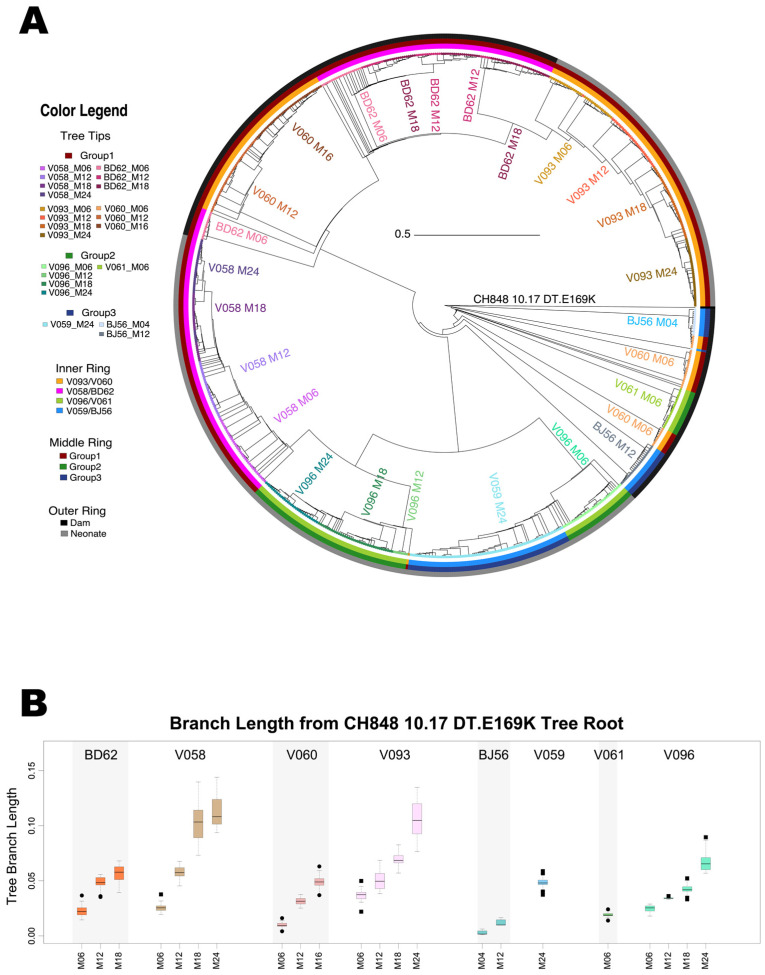
Phylogenetic reconstruction of HIV-1 Env sequences from SHIV-infected neonate and adult RMs. We studied 758 longitudinally sampled Env sequences from four neonates and four adult/dams. For the phylogenetic analysis, the Env sequences were rooted on the CH848.10.17 DT.E169K SHIV-infecting strain. (**A**) Filled circle tree tips indicate sequences from dam RMs, and squares represent sequences from neonate RMs. Warm colors are used to color code the dam and neonate pairs, BD62 and V058, and V060 and V093, respectively, where both the dams and neonates developed heterologous NAbs (Group 1). Shades of green are used for the pair V061/V096 (Group 2), where the neonate developed heterologous NAbs, but the paired dam did not. Shades of blue are used for the pair BJ56/V059 (Group 3), where neither the neonate nor the dam developed heterologous NAbs. Key for the colored bands forming the three rings outside the tree, from inside out: (i) paired animals; (ii) grouped animal pairs based on their heterologous HIV-1 NAb-developing profile; and (iii) adult (dark gray) or neonate (light gray) RM. The tree was constructed using FastTree software [26] from the amino acid sequence alignment using the JTT (Jones–Taylor–Thornton 1992) model and CAT approximation. (**B**) Boxplots of tree branch lengths, calculated from full tree shown in Figure 2A.

**Figure 3 viruses-16-01014-f003:**
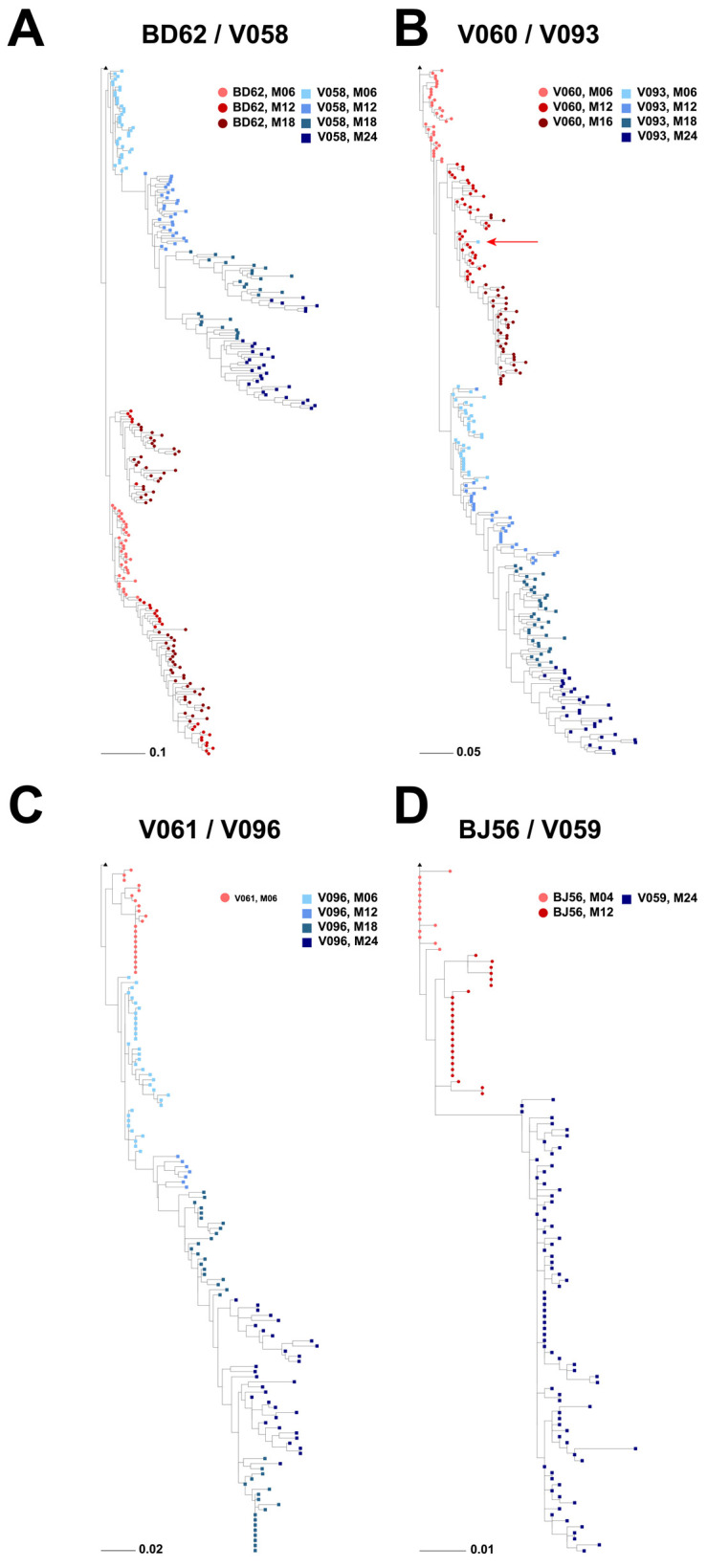
Phylogenetic trees depicting Env diversity in four neonate and adult/dam pairs of RMs. Each panel depicts the following dam and infant pairs: (**A**) dam-BD62, infant-V058; (**B**) dam-V060, infant-V093; (**C**) dam-V061, infant-V096; (**D**) dam-BJ56, infant-V059. Red hues, from lighter to darker, indicate longitudinal sequences from the dam RM, whereas blue hues, from lighter to darker, indicate sequences from the paired neonate RM. Trees are rooted on the CH848 1017.DT.E169K-infecting SHIV (black triangle). All trees were made using IQ-TREE software [27] with the HIVb model [28] and ultrafast bootstrap for branch support.

**Figure 4 viruses-16-01014-f004:**
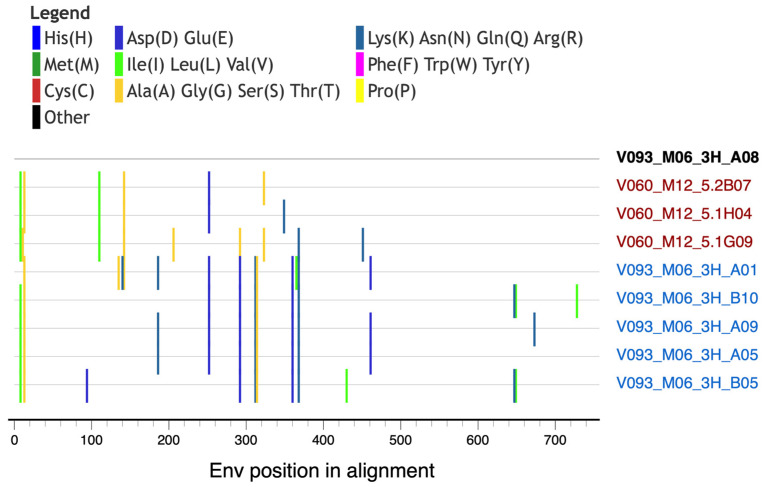
Sequence analysis of neonate V093 and corresponding dam V060 sequences. Highlighter plot showing outlier sequence V093_M06_3H_A08 compared to its within-host closest sequences (sequence names in blue) and to the closest sequence from the paired dam (sequence names in red). Amino acid mutations from V093_M06_3H_A08 are shown as color-coded tick marks.

**Figure 5 viruses-16-01014-f005:**
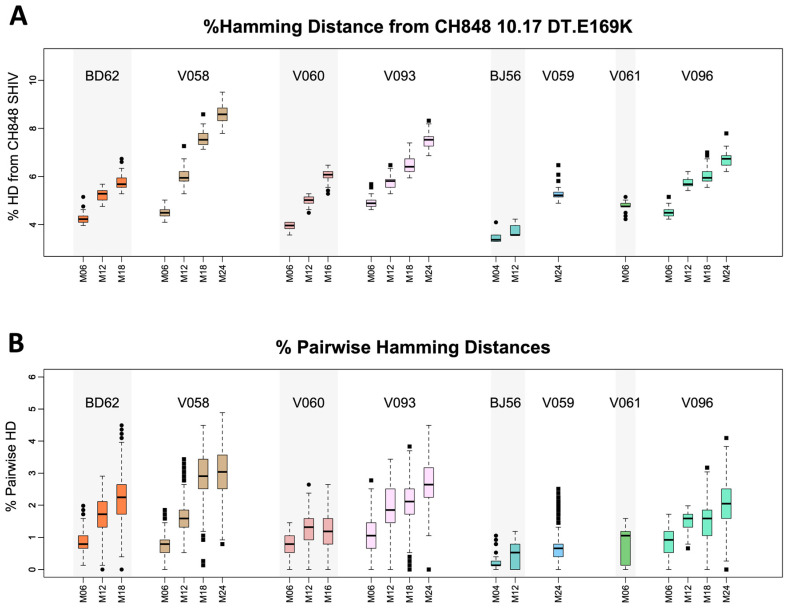
Env sequence diversity within host. Boxplots of within-time-point HD distributions from CH848 1017.DT.E169K (**A**), and within-time-point pairwise HD distributions (**B**) for each animal at each time point. HDs were calculated as follows: for panel (**A**), for each sequence, the HD was defined as the number of amino acid differences from CH848 1017.DT.E169K divided by the sequence length; for panel (**B**), for each sequence pair, the pairwise HD was defined as the number of amino acid differences between the two sequences divided by the sequence length. Among neonate RMs, animals that developed heterologous HIV-1 NAbs had the highest mean HDs at the M24 time point. The three RMs that did not develop heterologous HIV-1 NAbs had the three lowest mean HDs at all time points.

**Figure 6 viruses-16-01014-f006:**
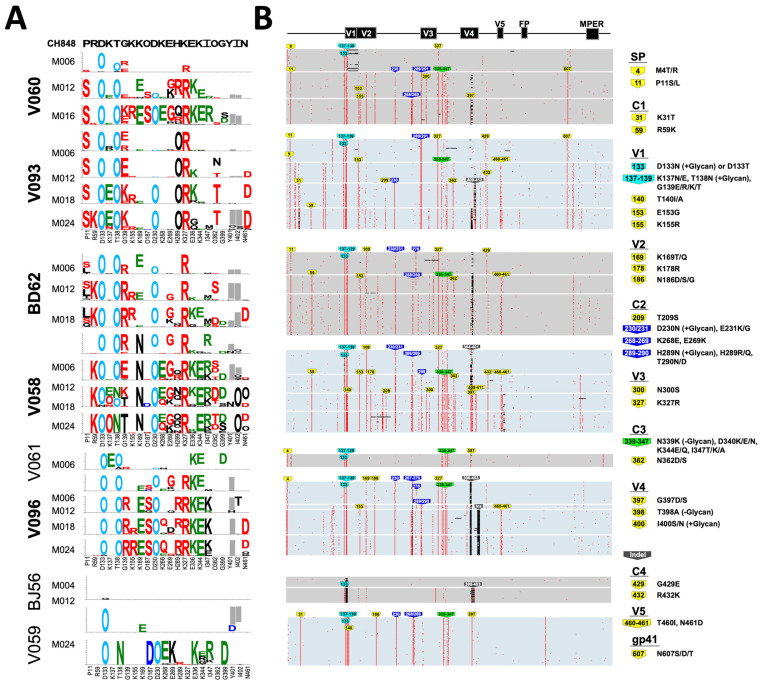
Env sequence diversification in RMs at the amino acid level. (**A**) Logo plots of 22 Env sites, where a 50% or higher loss of the CH848 1017.DT.E169K-infecting strain residue was observed in at least 3 animals; the sequence of the infecting CH848 1017.DT.E169K Env is shown at the top of the Logo plot. Logos are proportional to the frequency of the corresponding amino acid at each time point. “O” is used to indicate an asparagine embedded in a glycosylation motif. Amino acids at Env variable sites shared by 3 or more animals are color coded as follows: red, if they are found mostly in the HIV-1 heterologous NAb-developing RMs, blue if they are found in RMs that did not develop heterologous HIV-1 NAbs, green if they are mutating in both animal groups, and cyan if they are glycosylation gains in V1 (found in most animals). Gray rectangles indicate deletions. (**B**) Longitudinal HIV-1 Env sequence evolution of amino acid alignments from the four neonate and dam pairs visualized using a pixel plot. Gray boxes represent sequences from dam RMs and light blue boxes represent sequences from paired neonate RMs. Amino acid substitutions and insertions or deletions (indels) relative to the CH848 1017.DT.E169K-infecting sequence are colored red and black, respectively. Colored tags indicate amino acid positions (HXB2 numbering) where mutations away from the residue in the Env from the infecting strain are detected in ≥50% of the Env sequences in at least two RMs at any time point. Mutations in the V1 glycan hole, 241/289 glycan hole, and C3 region are indicated with cyan, blue, and green labels, respectively; all other mutations are labeled in yellow. Bold font indicates RMs that developed heterologous HIV-1 NAbs.

## Data Availability

The data supporting the findings of this study are available within the main article. Any further data relevant to the study will be made available upon request to the corresponding authors.

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
