# Peer review of "Viral Envelope Evolution in Simian–HIV-Infected Neonate and Adult-Dam Pairs of Rhesus Macaques"

_viruses, 2024, doi:10.3390/v16071014_

Round 1

Reviewer 1 Report

Comments and Suggestions for Authors

Overall: Elena E. Giorgi et al. expose complementary analysis from another study Holmes S, Williams WB et al. to present ENV evolution in SHIV infection from neonatal versus adult rhesus macaques. The authors found better heterologous HIV neutralizing antibody profile with a faster ENV diversity in neonate than in adult rhesus macaque.

Comments:

- While the study is dependent on another one, the introduction must be improved to better understand the aims and particularly the authors' hypothesis that is mentioned only in the discussion section.

- In the result section, several parts of the text are interpretation analyses, particularly for the figure 1, that is not described despite the group classification (i.e. line 136-141).

- There is a mistake at the line 165 about the V060 monkey.

- In figure 4, change the colors to have a better distinction of the amino acid.

- Line 256-257, V061 and BD62 is non-significatively different to V060 and BD62 at M06; compared M06 to M12 for different monkeys is not correct analysis.

- Line 359: Figure 6B

Author Response

REVIEWER #1

Overall: Elena E. Giorgi et al. expose complementary analysis from another study Holmes S, Williams WB et al. to present ENV evolution in SHIV infection from neonatal versus adult rhesus macaques. The authors found better heterologous HIV neutralizing antibody profile with a faster ENV diversity in neonate than in adult rhesus macaque.

Response: We thank the reviewer for the positive feedback and summary of our manuscript.

 Comments:

- While the study is dependent on another one, the introduction must be improved to better understand the aims and particularly the authors' hypothesis that is mentioned only in the discussion section.

Response: We thank the reviewer for the feedback. We have now modified the introduction (see lines 58-67) to introduce this concept earlier for a better understanding of the study design and outcomes. The introduction now addresses the concept outlined in now lines 357-359 of the discussion section of the manuscript where we stated that “Altogether, these data support the hypothesis that Env sequence diversity is a key driver for development of neutralization breadth (Wagh et al., Cell Rep., 2018; Roark et al., Science, 2020)”.

- In the result section, several parts of the text are interpretation analyses, particularly for the figure 1, that is not described despite the group classification (i.e. line 136-141).

Response: We thank the reviewer for the feedback and have modified the text in the results to avoid interpretation analyses and describe the data. However, in some instances, we include added text to provide the context for the purpose of the studies before describing the data, including lines 136-411 (now lines 147-150). Lines 157-162 is another example of the modified text to address this comment.

- There is a mistake at the line 165 about the V060 monkey.

Response: V060 and BD62 were the two dams that generated heterologous HIV-1 NAbs as outlined in line 165 and shown in Figures 1A-1B. Sequences from V060 M12 and M16 clustered with sequences from BD62 as shown in Figure 2A. We have now modified the text in lines 179-183 for clarity.

- In figure 4, change the colors to have a better distinction of the amino acid.

Response: We apologize to the reviewer if the figure is confusing. The figure is automatically generated by the LANL tool highlighter   (https://www.hiv.lanl.gov/content/sequence/HIGHLIGHT/highlighter_top.html) and by design it groups certain amino acids under the same colors because it’s really hard to find a palette of colors that would distinguish all of the 20 amino acid. However, the point of the figure is to highlight the number of differences/mutations from the reference sequence shown at the top. Therefore, even though some colors are similar, we feel that the figure still conveys the message of which sequences in the set are most similar and which are most diverse compared to V093_M06_3H_A08. 

- Line 256-257, V061 and BD62 is non-significatively different to V060 and BD62 at M06; compared M06 to M12 for different monkeys is not correct analysis.

Response: Lines 237-240 states that “we found that the two adult RMs that developed heterologous HIV-1 NAbs (V060 and BD62) had significantly higher median pairwise HD compared to the ones that did not (BJ56 and V061); all comparisons p<2x10-9 by Wilcoxon test at the latest common time point for which sequences were available – M12 for dams BD62, V060 and BJ56, but M06 for dam V061.” Assuming this is the statement the reviewer is referring to, then the reviewer is correct in that the comparison we are making is skewed by the fact that for animal V061 unfortunately we only had one time point available and that was M06. The whole study is based on the data we have. We were not able to sequence later time points for some of the adult animals due to very low viral loads, so we are restricted to make inference based on the time points for which we do have data. We have now added a sentence to acknowledge this limitation of our study in lines 346-347.

- Line 359: Figure 6B

Response: Line 359 does not reference a figure, but I am guessing that the reviewer was referring to line 265 (now line 285). See line 285 for the correction, Figure 6B.

Reviewer 2 Report

Comments and Suggestions for Authors

Brief summary Giorgi et al. characterize the genetic diversity of Env sequences from 4 neonate/adult-dam RM pairs. The paper starts by characterizing the neutralization profiles of the 4 neonate/adult-dam RM pairs to establish 3 'groups': (a) both neonate and dam develop plasma heterologous HIV-1 NAbs, (b) the neonate developed high titer plasma heterologous HIV-1 NAbs but not the dam (c) neither the neonate or the dam developed plasma heterologous NAbs. These groups set the foundation for the exploration of the mechanisms that underlie neonatal bnAb induction. The authors utilize various methods (phylogenetic reconstruction of HIV-1 Env sequences, Phylogenetic trees depicting Env diversity, Sequence analysis of Env, Env sequence diversity, etc.) to demonstrate that Env sequence diversity in a key driver for the development of neutralization breadth. 

General Comments:

1. In the results section of Figure 1 the authors explain that they grouped the 4 neonate/dam pairs using their plasma neutralization profile. What is not clear is what were the criteria to establish whether a neonate or dam had 'developed plasma heterologous HIV-1 NAbs'. Additionally, which timepoints are the most informative for this distinction? Without a sound explanation it is difficult for the reader to understand why these groups are analyzed downstream as such. 

2. I am assuming that these 4 neonate/dam pairs were pulled from the Holmes S, Williams SB et al paper in submission that was mentioned to have 11 neonate/dam pairs. What is not clear is why specifically these 4 pairs were pulled for this study. Where they the only pairs to induce plasma heterologous bnAbs?

3. Conceptually the results section of Figure 3 is sound. My main gripe with this section is figure 3 is illegible in it's current state making it hard to corroborate the data with the results description. 

4. Lines 230-232 state 'These data suggest... host specific immune pressures'. It is not clear how the authors are making this statement. The authors do not show any data that the neonate and dams have similar Envs after infection. The pair looked at in Figure 4 looks at the V093/V060 pair at two different timepoints M12 for V060 and M06 for V093. Nor does this figure show that Env diversify differently due to host specific immune pressures. In order to make this statement I would want to see (A) sequencing comparison analysis of neonate and dam at the same timepoint 6M or 12M  (B) a similar pattern via sequence analysis for the remaining neonate/dam pairs. 

The results section of Figure 5 would benefit from a brief explanation as to why pairwise hamming distances were included in the analysis. 

5. An introductory figure illustrating the course of infection and sampling timepoints should be included to simplify the study design to the reader.  

6. For Figure 6 I echo the same sentiments I had with Figure 3. Conceptually the results section is sound. My main gripe with this section is figure 6 is illegible in it's current state making it hard (if not impossible) to corroborate the data with the results description. 

Line comments:

Line 39 replace 'another' with 'other' & replace 'lineage' with 'lineages'  

Line 48 the use of 'gold standard' here is not appropriate. 

Line 50 redact 'and'

Line 51 revise 'mature members of the B cell lineage to bnAb status'. Unsure what this means in the context of bnAb vaccine design. 

Line 51-56 revise 'The success of this vaccine.... bnAb lineage maturation.' This is a run on sentence with a lot of ideas. Consider splitting into 2 sentences. 

Figure Comments:

Figure 1.

Figure should be at a higher resolution as current figure is quite blurry 

The far right of the figure seems to be partially cropped.

Figure 2A.

Legend for 0.5 line is missing

Color legend and other text is illegible at 100% 

Recommend to not use light colors (cyan for example) as they are difficult to read on a white background

Figure should be at a higher resolution as current figure is quite blurry and difficult to read

Figure 2B.

Figure should be at a higher resolution as current figure is quite blurry and difficult to read

X and Y axis labels are currently illegible 

Figure 3. 

Figure should be at a higher resolution as current figure is quite blurry and difficult to read

Hard to visualize currently, the figure should be presented at a larger scale for easier visualization 

Legend for 0.1,0.05, etc. line is missing

Figure 4. 

Figure should be at a higher resolution as current figure is blurry 

Figure 5.

X and Y axis text is currently too small and illegible

Figure should be at a higher resolution as current figure is blurry 

Hard to visualize currently, the figure should be presented at a larger scale for easier visualization 

Figure 6. 

Figure should be at a higher resolution as current figure is blurry 

Hard to visualize currently, the figure should be presented at a larger scale for easier visualization 

Author Response

REVIEWER #2

Brief summary:  Giorgi et al. characterize the genetic diversity of Env sequences from 4 neonate/adult-dam RM pairs. The paper starts by characterizing the neutralization profiles of the 4 neonate/adult-dam RM pairs to establish 3 'groups': (a) both neonate and dam develop plasma heterologous HIV-1 NAbs, (b) the neonate developed high titer plasma heterologous HIV-1 NAbs but not the dam (c) neither the neonate or the dam developed plasma heterologous NAbs. These groups set the foundation for the exploration of the mechanisms that underlie neonatal bnAb induction. The authors utilize various methods (phylogenetic reconstruction of HIV-1 Env sequences, Phylogenetic trees depicting Env diversity, Sequence analysis of Env, Env sequence diversity, etc.) to demonstrate that Env sequence diversity in a key driver for the development of neutralization breadth. 

Response: We thank the reviewer for the positive feedback and summary of our manuscript.

General Comments:

  1. In the results section of Figure 1 the authors explain that they grouped the 4 neonate/dam pairs using their plasma neutralization profile. What is not clear is what were the criteria to establish whether a neonate or dam had 'developed plasma heterologous HIV-1 NAbs'. Additionally, which timepoints are the most informative for this distinction? Without a sound explanation it is difficult for the reader to understand why these groups are analyzed downstream as such. 

Response: The revised sentences on lines 147-150 and 157-162 now clarified the criteria for heterologous HIV-1 neutralization that we used to define bnAb induction. These criteria was also previously defined in Hora et al. (Cell Reports, 2023) and referenced in this manuscript. Another manuscript (Holmes S, Williams WB et al.) is currently under review at an independent journal, where we described the total of 11 neonate/dam pairs for comparison of virologic and immunologic responses to the same SHIV. The 4 pairs of dam/neonate pairs described in this manuscript reflect the 2 pairs of dams/neonates (group 1) where heterologous HIV-1 NAb induction was described, whereas V096/V061 neonate/dam pair was representative of 5 neonate/dam pairs where only the neonate generated heterologous HIV-1 NAbs, and V059/BJ56 was representative of 4 neonate/dam pairs where neither neonate nor dam generated heterologous HIV-1 NAbs. The representative neonate/dam pairs had the most viral Env sequences recovered from longitudinal time points. We now provide this information that further describes the criteria for the animals in this study on lines 103-111 of the methods.

  1. I am assuming that these 4 neonate/dam pairs were pulled from the Holmes S, Williams SB et al paper in submission that was mentioned to have 11 neonate/dam pairs. What is not clear is why specifically these 4 pairs were pulled for this study. Where they the only pairs to induce plasma heterologous bnAbs?

Response: See the response above and lines 97-111 of the methods.

  1. Conceptually the results section of Figure 3 is sound. My main gripe with this section is figure 3 is illegible in it's current state making it hard to corroborate the data with the results description. 

Response: We thank the reviewer for the feedback. Now lines 201-207 describe these data which shows phylogenetic trees where sequences are showed as color symbols instead of sequence ID. We are unsure if our image has been modified after submission, but the trees appeared very clear to us. However, we will put all text in bold and generate a more high-resolution version of this image for the resubmission.

  1. Lines 230-232 state 'These data suggest... host specific immune pressures'. It is not clear how the authors are making this statement. The authors do not show any data that the neonate and dams have similar Envs after infection. The pair looked at in Figure 4 looks at the V093/V060 pair at two different timepoints M12 for V060 and M06 for V093. Nor does this figure show that Env diversify differently due to host specific immune pressures. In order to make this statement I would want to see (A) sequencing comparison analysis of neonate and dam at the same timepoint 6M or 12M  (B) a similar pattern via sequence analysis for the remaining neonate/dam pairs. 

Response: This conclusion now stated on lines 214-216. All animals were challenged and infected with the same sequence strain, SHIV CH848 10.17 N133DN138T.E169K, so the expectation that shortly after infection env sequences across animals are similar, as they evolve from the same founder lineage, is not unreasonable. Figure 3B showed the clustering of a single V093 (neonate) M06 sequence with V060 (dam) M12 sequences, and Figure 4 provided a summary of the analysis of the sequences at these time points in these two animals. Viral envs then evolve in each animal according to the immune pressure they are subjected to in each individual host. However, the fact that in this infant in particular one env clustered closer to the maternal sequences was suggestive of the initial immune pressures from the hosts being similar and then diverged later on as the majority of the infant’s virus env sequences formed a distinct subclade. Lines 212-214 indicated that these samples were not prepared simultaneously thus limiting the likelihood of sample contamination. We have revised lines 214-216 to be more cautious and indicate that our analyses “raised the hypothesis that both neonate and dam pairs have similar Envs soon after infection that may diversify differently due to host specific immune pressures”.

The results section of Figure 5 would benefit from a brief explanation as to why pairwise hamming distances were included in the analysis. 

Response: We have added text to lines from 221-230 for clarity. While the two measures of diversification we are describing in the text tend to be correlated, we felt that distances from the infecting strain alone would not capture the full evolutionary history as in some instances multiple quasi-species branch out of a single initial lineage as the virus explores distinct pathways of escape from the host’s immune pressure.

  1. An introductory figure illustrating the course of infection and sampling timepoints should be included to simplify the study design to the reader.  

Response: Lines 100-103 describes the course of infection for the animals in this study and references the papers that described the sampling timepoints, including the companion paper that more extensively describes the virologic and immunologic responses to the same SHIV in all the pairs of neonate and dams in our cohort, from which we studied 4 pairs as outlined above. To avoid overlap, we prefer to reference the paper with the introductory figure that is under review. Given that this is a solicited article for a special issue in Viruses, we can also discuss further with the editors to wait until the companion paper is published or we can provide a copy of that paper for their review.

  1. For Figure 6 I echo the same sentiments I had with Figure 3. Conceptually the results section is sound. My main gripe with this section is figure 6 is illegible in it's current state making it hard (if not impossible) to corroborate the data with the results description. 

Response: We are unsure if our image has been modified after submission, but Figure 6 appeared very clear to us in the pdf that was submitted and none of the other reviewers provided a critique of the image quality. We will try to improve resolution if possible prior to resubmission.

Line comments:

Line 39 replace 'another' with 'other' & replace 'lineage' with 'lineages'  

Response: Now lines 46-47 have been edited.

Line 48 the use of 'gold standard' here is not appropriate. 

Response: Line 55 has now been revised to replace “gold standard” with “a goal”.

Line 50 redact 'and'

Response: Now line 57. “And” redacted.

Line 51 revise 'mature members of the B cell lineage to bnAb status'. Unsure what this means in the context of bnAb vaccine design. 

Response: Line 57 is now revised for clarity and mentions that “boosting immunogens mature members of the B cell lineage to bnAb status”. BnAb status with heterologous HIV-1 neutralization activity is acquired at different stages of bnAb lineage maturation going from the non-neutralizing precursors to mature lineage members with neutralization capacity (ie, bnAbs). These concepts may be more clearly described on revised lines 58-63.  

Line 51-56 revise 'The success of this vaccine.... bnAb lineage maturation.' This is a run on sentence with a lot of ideas. Consider splitting into 2 sentences. 

Response: Now lines 63-67 has been revised for clarity.

Figure Comments:

Figure 1.

Figure should be at a higher resolution as current figure is quite blurry 

The far right of the figure seems to be partially cropped.

Response: We apologize for any problems with the image that was provided to this reviewer by the journal, but we will discuss further with the handling editor. It may be a problem with the files provided to this reviewer, but the pdf of figure 1 provides a high-resolution image that is not blurry nor cropped.

Figure 2A.

Legend for 0.5 line is missing

Color legend and other text is illegible at 100% 

Recommend to not use light colors (cyan for example) as they are difficult to read on a white background

Figure should be at a higher resolution as current figure is quite blurry and difficult to read

Response: See above. We apologize for any problems with the image that was provided to this reviewer by the journal, but we will discuss further with the handling editor. This problem appears to be consistent with all the figures for this reviewer. Thus, we will discuss with the editors to provide the figures as a different file format that may not be affected when opened by this reviewer. Figure 2A and 2B were very clear in the pdfs submitted and neither blurry nor hard to read, despite the color choices of the legend.

Figure 2B.

Figure should be at a higher resolution as current figure is quite blurry and difficult to read

X and Y axis labels are currently illegible 

Response: See responses above. We apologize for any issue that may have occurred with the images provided to this reviewer.

Figure 3. 

Figure should be at a higher resolution as current figure is quite blurry and difficult to read

Hard to visualize currently, the figure should be presented at a larger scale for easier visualization 

Legend for 0.1,0.05, etc. line is missing

Response: See responses above. We apologize for any issue that may have occurred with the images provided to this reviewer. The legends were visible in the pdfs of the images submitted and the image was not blurry and quite clear.

Figure 4. 

Figure should be at a higher resolution as current figure is blurry 

Response: See responses above. We apologize for any issue that may have occurred with the images provided to this reviewer. Figure 4 submitted as a pdf did not have the issues raised by this reviewer.

Figure 5.

X and Y axis text is currently too small and illegible

Figure should be at a higher resolution as current figure is blurry 

Hard to visualize currently, the figure should be presented at a larger scale for easier visualization 

Response: See responses above. We apologize for any issue that may have occurred with the images provided to this reviewer. We will discuss further with the journal to avoid this problem with the resubmitted documents.

Figure 6. 

Figure should be at a higher resolution as current figure is blurry 

Hard to visualize currently, the figure should be presented at a larger scale for easier visualization 

Response: See responses above. We apologize for any issue that may have occurred with the images provided to this reviewer. We will discuss further with the journal to avoid this problem with the resubmitted documents.

Reviewer 3 Report

Comments and Suggestions for Authors

Author Response

REVIEWER #3

we characterized 15 genetic diversity in Env sequences from 4 neonate/ adult-dam RM pairs with different phenotypic profiles of developing bNABs. Previously characterized the breadth of circulating plasma antibodies in each neonate/dam pairing. Picked 4 to study. Seuqeneced Env and characterized genetic evolution. Figure 2 seems redundant to Figures 3 and 5? Not sure what it is showing that isn’t more clear in the pair-specific figures.

Response: We thank the reviewer for the feedback. Both Figures 2 and 3 present a visualization of the phylogenetic analyses of the Env sequences in the animals studied. Phylogenetic reconstructions change depending on the input sequence set as they show the evolutionary history that’s shared across all sequences and hence how the sequences all related to one another. Therefore, Figure 2, because it is a phylogenetic reconstruction of all sequences across all animals, shows how all the sequences compare across total neonates and dams studies. For example, it shows how env sequences tend to cluster by age rather than by host pair. But we also wanted to see whether within each pair there would be interesting evolutionary patterned that were shared between the dam and the neonate, and this is why we show figure two with the phylogenetic tree for each individual pair. In both analyses, we described important findings that were more evident in the independent analyses performed and visualized in different figures.

Figure 5 shows hamming distance, which are correlated to the phylogenetic branch lengths shown in figure 2B, but not always. We had to present figure 5 as well to make the point that they are correlated, one cannot make the assumption at priori.

For Figure 2B, can which NHP develop het neut be indicated?

Response: We thank the reviewer for this feedback. This figure was already complicated with a lot of different colors, thus we decided to indicate the NHPs that developed heterologous HIV neutralization activity in the figure legend. See lines 413-417 that outlined a key for this figure.

Figure 3 resolution was very poor and impossible to see the black triangle indicating the T/F. It is not clear from Figure 3A that the dam BD62 sequences are closer to the T/F than sequences from V058 as said on LN 214 Figure 6 was also really hard to read.

Response: We apologize for any inconvenience with the images provided to the reviewer. The images submitted to journal as pdfs were high-resolution and clear. We will make all attempts to improve resolution and ensure that the files received by the journal are of the best quality.

Round 2

Reviewer 1 Report

Comments and Suggestions for Authors

Reviewer 2 Report

Comments and Suggestions for Authors

Thank you for addressing my comments.